# Peer review of "Pretreatment Radiologically Enlarged Lymph Nodes as a Significant Prognostic Factor in Clinical Stage IIB Cervical Cancer: Evidence from a Taiwanese Tertiary Care Center in Reaching Consensus"

_diagnostics, 2022, doi:10.3390/diagnostics12051230_

Round 1

Reviewer 1 Report

The manuscript entitled “Pretreatment Radiologically Enlarged Lymph Nodes as a Significant Prognostic Factor in Clinical Stage IIB Cervical Cancer: Evidence from a Taiwanese Tertiary Care Center in Reaching Consensus” analyzed the prognostic significance of pretreatment radiologic LN status in patients with clinical stage IIB CC. This is an interesting manuscript and well-written, but I am concerned about some statistical analyses. The Authors used the student’s t-test to analyze continuous variables. However, this is a parametric test that should only be applied to data with normal distribution. Were the data normality tested? With which test?

Author Response

Reviewer #1

The manuscript entitled “Pretreatment Radiologically Enlarged Lymph Nodes as a Significant Prognostic Factor in Clinical Stage IIB Cervical Cancer: Evidence from a Taiwanese Tertiary Care Center in Reaching Consensus” analyzed the prognostic significance of pretreatment radiologic LN status in patients with clinical stage IIB CC. This is an interesting manuscript and well-written, but I am concerned about some statistical analyses. The Authors used the student’s t-test to analyze continuous variables. However, this is a parametric test that should only be applied to data with normal distribution. Were the data normality tested? With which test?

Response: Thank you for your valuable comments. In this study, Shapiro-Wilk's test was used to test the normality of the continuous variables before performing the student's t-test. The results of Shapiro-Wilk's test showed that all continuous variables are not significantly different from the normal distribution (all p-values > 0.05). The normality test result has been provided below in (Table S3).

Table S3. Shapiro-Wilk’s test shows the normal distribution for all continuous variables.

Age w-value

Size w-value

Age p-value

Size p-value

Total

0.985

0.981

0.5524

0.501

LN

0.985

0.969

0.9176

0.585

non-LN

0.948

0.972

0.0682

0.542

Based on your comment, we have now uploaded Table S3 as supplementary data.

Reviewer 2 Report

The design of the study is proper, the number of the patients is large enough to provide a powerful statistic evaluation. The conclusions are clear, they are supported by the statistics and are also sustained by the literature citations found in the article.

Reviewer 3 Report

The work is very interesting and well written. I think it would be important to extend the topic to a larger patient population and to other countries. Because this work focuses on the analysis of 72 cases, and half of them were diagnosed with enlarged lymph nodes (a small number of people).

            I have a few minor comments:

  1. On page 4, line 134 is: “(Table 2)” and should be: “(Table 2)” (bold);
  2. On page 6 is Figure 1 that is not in the text (didn't I notice?);
  3. On page 6, there should be a space between lines 172 and 173;
  4. On page 7, line 182 is: “in Table 4”, and should be: “in Table 4” (bold);
  5. On page 7, there should be a space between lines 184 and 185;
  6. On page 7, line 202 is: “Schneidler et al.” and it should be: “Scheidler et al.”;

Author Response

Reviewer #2

The work is very interesting and well-written. I think it would be important to extend the topic to a larger patient population and to other countries. Because this work focuses on the analysis of 72 cases, and half of them were diagnosed with enlarged lymph nodes (a small number of people).

Response: Thank you for appreciating our work and your valuable comments. We totally agree that it is important to further extend our work to a larger scale based on our research findings. In fact, we are planning to extend our work to analyze national-wise cancer registry databases in Taiwan after gaining permission to access the national-wise databases maintained by Taiwan’s Ministry of Health. We have revised the manuscript and stressed that a larger population-based study is necessary to confirm our findings as follows:

Page 8, line no. 256-258:

“Despite the inherent limitations, the data presented in the current study is worthy of further investigation, and a larger population-based study is necessary to confirm our findings.”

Minor comments:

  1. On page 4, line 134 is: “(Table 2)” and should be: “(Table 2)” (bold).

Response: Thank you for your suggestion. Based on this comment, we have changed (Table 2) to (Table 2)accordingly and revised the manuscript as follows:

Page 4, lines 134-135:

 “Univariate Cox analysis (Table 2) revealed that recurrence was directly influenced by LN metastatic level according to Querleu and Morrow.”

  1. On page 6 is Figure 1 that is not in the text (didn't I notice?).

Response: Thanks for the reminder. We have now mentioned the Figure 1A and Figure 1B in the revised manuscript as follows:

Page 4, line no. 126-131:

“Specifically, the 1-, 2-, and 5-year PFS rates (95% CI) were 69.7% (51.0–82.4), 63.5% (44.7–77.4), and 47.0% (27.9–64.0) for the LN group; and 94.8% (80.8–98.7), 89.5% (74.5–95.9), and 72.6% (53.1–85.0) for the non-LN group, respectively (Figure 1A). OS also exhibited a similar trend. The 1-, 2-, and 5-year OS rates were 78.8% (60.6–89.3), 66.7% (47.9–80.0), and 38.1% (20.9–55.2) for the LN group; and 97.4% (83.2–99.6), 92.1% (77.4–97.4), and 76.7% (58.3–87.8) for the non-LN group, respectively (Figure 1B).”

  1. On page 6, there should be a space between lines 172 and 173.

Response: Thank you for your suggestion. Based on this comment, we have added a space between lines 172 and 173 accordingly and revised the manuscript.

  1. On page 7, line 182 is: “in Table 4”, and should be: “in Table 4” (bold).

Response: Thank you for your suggestion. Based on this comment, we have changed “in Table 4” to “in Table 4(bold) and revised the manuscript as follows:

Page 7, lines 187-188:

“The frequency of regional LN metastasis in stage IIB CC reported in previous studies is presented in Table 4.

  1. On page 7, there should be a space between lines 184 and 185.

Response: Thank you for your valuable comments. We have added a space between lines 184 and 185 and revised the manuscript accordingly.

  1. On page 7, line 202 is: “Schneidler et al.” and it should be: “Scheidler et al.”

Response: Thank you for the valuable comment. Based on this comment, we have corrected the spelling error and revised the manuscript as follows:

Page 7, lines 208-209:

“Notably, a meta-analysis by Scheidler et al. reported that CT had a positive predictive value of 61% during pretreatment evaluation for the diagnosis of LN metastasis [30].”
